# Process-Oriented Design Methodologies Inspired by Tropical Plants

**Elizabeth L. McCormick** [1,2,*], **Elizabeth A. Cooper** [3,4], **Mahsa Esfandiar** [1], **MaryGrayson Roberts** [1] **and Lindsay Shields** [3,4]

1. School of Architecture, University of North Carolina at Charlotte, Charlotte, NC 28262, USA
2. College of Design, North Carolina State University, Raleigh, NC 27607, USA
3. Department of Bioinformatics and Genomics, University of North Carolina at Charlotte, Charlotte, NC 28223, USA; ecoope23@uncc.edu (E.A.C.)
4. North Carolina Research Campus, Kannapolis, NC 28081, USA
* Correspondence: emccorm4@charlotte.edu; Tel.: +1-704-687-0111

**Abstract:** In light of the escalating climate crisis, there is a pressing need for a significant shift in how we design the built environment to effectively confront global challenges. Natural systems have inspired scientists, architects, and engineers for centuries; however, conventional biomimetic approaches often focus on superficial aspects, disregarding the underlying complexities. While this approach may lead to a more efficient outcome, it operates under the assumption that the organism functions exclusively within the confines of human knowledge, which are inherently limited by established epistemological and technological systems. This study advocates for a departure from conventional biomimetic approaches and asks the mechanisms of the biological system to inform the *process* of translation, as opposed to simply defining the *outcome*. By relinquishing control to material properties and dynamic processes of the biological analog, this study explores the generation of novel, bio-inspired dynamic formworks through non-linear fabrication processes. Specifically, it investigates the thermal properties of accessible building materials, enabling them to respond to environmental conditions without sophisticated technology or human intervention. By embracing chance and unpredictability, translated behaviors are granted the same influence as human intervention. Drawing inspiration from adaptive plant physiology, this research seeks to inspire innovative, climate-responsive methodological practices within broader architectural systems.

**Keywords:** biomimicry; dynamic formwork; adaptive facades; climate-responsive

## 1. Introduction

Since the industrial revolution, the building and construction industry has experienced several major shifts, including mechanization and mass production (late 18th century), reinforced concrete skyscrapers (late 19th century), as well as computer-aided design (CAD) and building information modeling (BIM) (late 20th century), to name a few. While some of these developments have had a dramatic influence on the performance and resiliency of the built environment, the building industry is still responsible for 30% of global final energy consumption and 26% of global energy-related emissions, according to the International Energy Agency (IEA). Despite improvements in advanced technologies, energy efficiency, and increasingly rigorous energy codes, building sector energy consumption continues to increase [1]. This evidence supports the need for a substantial recalibration of the way we design and construct the built environment to better address global changes. The fusion of technological advancements with lessons from nature could hold the key to achieving a harmonious balance between human infrastructure and the environment. In doing so, however, we must discover how to truly learn from nature.

Natural systems have inspired scientists, architects, and engineers for centuries. This practice, now known as biomimicry, aims to translate natural solutions into novel, sustainable human technologies. Many biomimetic approaches, however, explore only the superficial characteristics of the natural analog, including anatomical structure, shape, and texture, often overlooking the intricate underlying mechanisms that underpin the complexities of the biological model. While this approach may lead to more efficient outcomes, it implicitly assumes that the organism operates solely within the boundaries of human knowledge, which are inherently limited by established epistemological and technological systems. While a deeper understanding of the natural analog can lead to more advanced technologies, it does not challenge the fundamental premise that our understanding of an organism exists only within the framework of human knowledge and understanding. Instead, this research seeks to challenge traditional biomimetic design practices and construction norms in pursuit of more adaptive and resilient solutions for sustainable architecture.

This paper will first outline a brief survey of relevant literature before describing in detail the biological system used in translation. Next, it will describe a methodological prototype inspired by the adaptive physiological behaviors found in tropical plants, grasses, and sedges. The goal of this research is not just to spark creative architectural ideas based on natural systems, but to establish useful methods based on the evolutionary and adaptive traits of biological systems. Specifically, this research explores the dynamic functions of plants to inform dynamic processes of construction, without focusing on a predetermined outcome.

## 2. Background

### 2.1. Biomimicry as a Field of Inquiry

In its broadest definition, biomimicry is an all-encompassing approach to learning from nature. Janine Benyus is the biologist credited with the popularization of biomimicry as a method of innovation. In her canonical text, *Biomimicry: Innovation Inspired by Nature* (1997), she articulates the role of nature in three conditions: nature as model, nature as measure and nature as mentor. Benyus states that natural systems can serve as a model by analyzing the forms, processes, systems, and strategies to answer human questions. As mentor, nature provides the rules to live by, learning from nature instead of exploiting it. Finally, as measure, nature provides the standards by which to judge the performance of human innovations [2]. Though she describes the ways that one can learn from nature, environmental philosopher Henry Dicks suggests that Benyus does not go far enough to define what nature is, which is key to the successful mimicry of nature [3].

This ambiguity, and lack of systematic design methodology, has allowed for a variety of different approaches to biomimicry as a field of inquiry, which leaves the research model "philosophically under-developed" [4] and limits the researcher's ability to imitate the entire natural process as a system [5,6]. Without a background in the biological sciences, engineers and designers may have a hard time deciphering and identifying which system in nature is worth abstracting and applying in practice, casually blurring the lines between morphology (form) and physiology (function).

Two prominent biomimetic architectural projects are SOMA's One Ocean Thematic Pavilion in South Korea (2012) and Mick Pearce's Eastgate Centre in Zimbabwe (1996). The One Ocean Project used a kinetic facade to replicate gills, the organ used by fish to exchange oxygen and breathe under water. The project uses metal louvers to frame views and block solar gains throughout the day, responding to the physical appearance of the gills. Pearce's Eastgate Centre, on the other hand, uses thermal mass and stack ventilation to replicate the thermoregulating qualities of an African termite mound. While this approach does reduce the need for air-conditioning systems in the temperate Zimbabwean climate, it neglects the physiological nuances developed by the species. In reality, the form of the termite mound is actually shaped to cultivate fungi, which create the buoyancy flow used to ventilate the subterranean colony (in tandem with the stack effect). Without consideration for this critical

fact, the Eastgate Centre uses electronically controlled fans to create the pressure differential needed to passively cool the building [7]. While both projects are prominent examples of bioinspired design, they do not holistically address the physiological processes of the natural analogs, focusing on form generation over function and behavior. When exclusively motivated by an organism's morphological attributes, it is likely that the designer will inadequately address the organism's capacity to adapt to severe or adverse conditions. Consequentially, in pursuit of promoting designs that operationalize natural processes, the holistic consideration of the full system becomes imperative.

López et al. (2017) recognized that biomimetic research was leaning more towards morphological translations and sought to provide a baseline methodology to aid other researchers in finding the correct method of biological translation [8]. Their research explored two corresponding design strategies to compare passive and dynamic translation strategies. First, they looked at seeds that are launched from their capsules for dispersal by a valve mechanism triggered by rainwater, to design a smart opening-closing system that can be used for new buildings (dynamic approach). Next, they explored another plant species with a reflective leaf structure to protect itself against excessive sunlight and temperatures to create a reflective envelope that can be applied to existing façades (static approach). In this process, they expressed the possibilities of biological processes to be applied in architecture that are not purely morphological. One project that focused on the physiological application with plants comes from Wiebe (2009), who draws connections between the chemical processes of photosynthesis with the thermal systems of the built environment [9]. In Wiebe's research, she found that the physiological development of Crassulacean Acid Metabolism (CAM), which evolved as a pathway for extreme conditions (high temperatures, low resources), could inform protective building enclosure systems. By comparing the chemical behavior of the biological model to the thermodynamic behavior of the building system, she developed a layered thermal wall with a variable barrier layer (thermochromic glazing) that provided the thermodynamic equivalent to the CAM pathway within the cell.

Biomimicry has propelled innovative design in many disciplines by learning from nature. However, challenges remain in comprehensively translating intricate natural systems and in establishing holistic methodologies that encompass the breadth of nature's adaptive strategies. To address this, an integrated approach is essential—one that not only recognizes the aesthetic qualities of natural forms but also responds to the underlying physiological conditions, fostering a symbiotic relationship between design and the natural world. Specifically, studying natural systems and their evolutionary adaptations provides invaluable insights for creating a more resilient and climate-responsive built environment. By channeling the resilience, adaptability, and resource-efficiency exhibited by plant life, architects and researchers are poised to create innovative and resilient design interventions tailored to the dynamic challenges of a rapidly changing climate. The next section will describe the plant model utilized by the authors for architectural translation and development of adaptive, climate-responsive design methodologies.

### 2.2. Climate-Responsive Materials

Compounding the issue of a holistic approach to biological translation, it is important to understand the inherent characteristics of the translation medium. For example, a thin piece of metal will have inherently different behaviors than a poured material, such as concrete. For that reason, it is important to balance the material integrity with the translation of the biological logic. Critiques of biomimetic practices have been met with a promising wave of emerging models that represent the successful integration of biological systems and material integrity. One example is the Hygroskin Meterosensitive Pavilion, shown in Figure 1, which is a 'climate-responsive' pavilion developed by architects Achim Menges, Oliver David Krieg and Steffen Reichert at the Institute for Computational Design (ICD) (2013). While many contemporary climate-responsive designs are enabled with mechanical and electronic sensing technologies, Hygroskin uses the responsive capacity ingrained in

the wood material itself. Inspired by the spruce pinecone which opens when dried and closes when wet, the pavilion apertures respond to the relative humidity changes from bright sunny to rainy weather in a moderate climate. In direct feedback to the microclimate, the pavilion constantly adjusts its degree of openness and porosity, modulating the light transmission and visual permeability of the envelope. This exchange results in constant fluctuations of enclosure, illumination and interiority of the internal space. The changing surface embodies the capacity to sense, actuate and react, due to the inherent qualities of the material itself. Similarly, Reichert et al.'s (2015) [10] approach uses hygroscopic (water-loving) materials that respond to changes in the environment in a way that mimics the way plants absorb moisture. Some researchers are even looking to alter the chemical properties of natural materials, such as wood, to enhance their thermo-regulating characteristics without the addition of mechanical attachments [11]. This process allows the materials to have an essential decision-making role in the final outcome of the translation, which helps to bridge the gap between human, material, and natural knowledge systems.

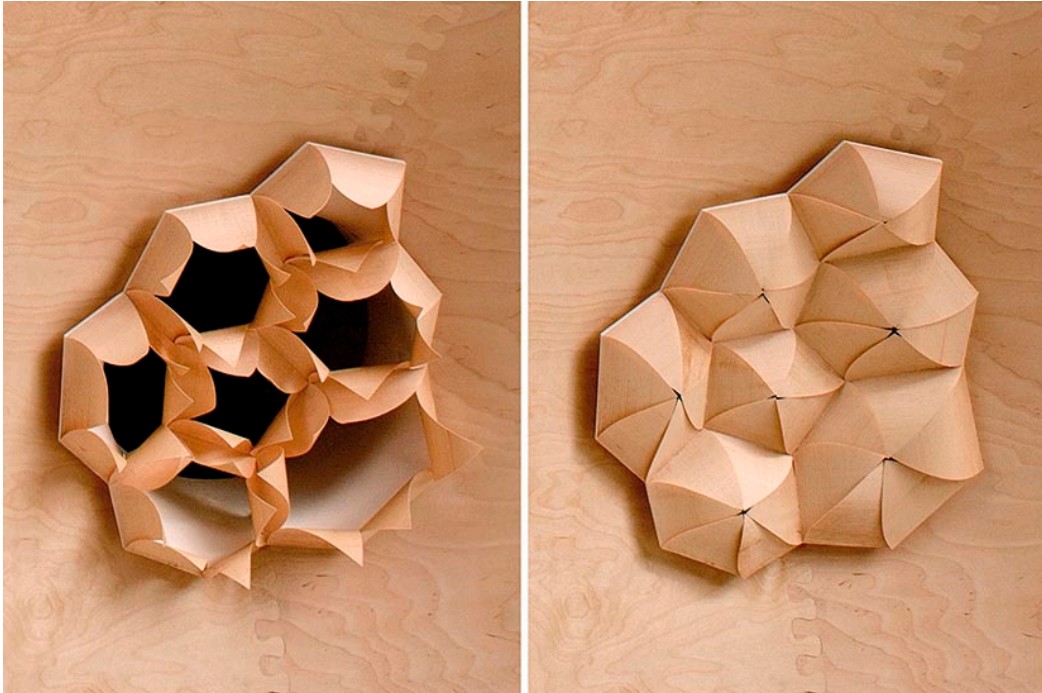

**Figure 1.** Hygroskin Pavilion, close-up of aperture adapting to weather changes: open at low relative humidity (**left**) and closed at high relative humidity (**right**). Images ©ICD/ITKE University of Stuttgart.

More 'advanced' materials with variable thermo-physical properties, are being used to maximize climate resources, particularly the natural fluctuations in solar energy and ambient air temperature [12]. Phase change materials (PCMs), for example, are increasingly used within buildings to buffer swings in temperature or humidity. These materials, through their capacity to absorb and release heat energy during phase transitions, provide an innate means of thermal regulation. Additionally, research such as Vazquez and Shaffer (2018) emphasize the significance of prioritizing *the process* over *the outcome* to promote novel and unforeseen outcomes. Their approach combines precision (robotics) with indeterminate processes (casting) of fabrication to challenge the geometrical control inherent in digital-material output and promote discovery over structured order [13]. This approach embraces opportunities for chance, error, unpredictability, and unexpected outcomes. Collectively, these studies underscore the potential of integrating natural principles, in tandem with innate material qualities to establish innovative and adaptable architectural systems, ultimately steering the field toward more responsive and sustainable systems.

### 2.3. Dynamic Formwork

Formwork serves as a temporary support during the construction of poured materials, such as concrete, and is primarily utilized to shape and preserve a desired geometry until the poured material has sufficiently cured to a level sufficient to support its own weight. Its origins trace back to the Roman Empire, where wooden shuttering was utilized by engineers to construct concrete vaults and arches. During the Renaissance period, the use of formwork increased to support the growth of concrete buildings, offering a more efficient alternative to stone masonry construction. As concrete usage continued to grow in the 20th century, numerous types of formwork emerged [14]. Complex concrete structures, particularly those with non-orthogonal geometries, can incur significant construction costs. This challenge is not exclusive to the construction industry as the marine, aerospace, automotive, transportation, and energy sectors also face similar situations where specialized shapes with low repetition rates must be manufactured, leading to high tooling costs due to low quantities, required accuracy, and high variability [15]. Although prefabrication can help reduce these costs, this approach often relies on repetitive elements to be economically viable. Instead, a reconfigurable mold is a more favorable option for small-scale production or rapid prototyping. In recent years, the use of robot-milled materials, such as polystyrene, fiberglass, plastics, and others have been used to accommodate complex geometries.

Oesterle et al. (2012), for example, proposed a waste-free, "free-form formwork", for making concrete formwork from machined wax in a way that is free-form, cast on-site, and not repetitive. Once the wax mold has cured, it can be assembled on standard scaffolding and used for casting concrete. After the formwork is removed, the wax pieces can be melted down and reused to make new forms [16]. Similarly, Schipper and Grünewald (2014) used advanced flexible formworks to help reduce the material waste associated with CNC-milled formwork systems [17]. Instead, they use thin and structurally efficient systems reinforced by textiles or fibers to produce thin, double-curved building shapes. Ice has also emerged as a recyclable, cost-effective, and environmentally friendly material in fabrication processes. Sitnikov (2019) and Sitnikov et al. (2019) propose the use of frozen aggregate as a base material for CNC-milled formwork, to create complex concrete shapes and reduce the environmental impact associated with conventional formwork materials [18,19]. The ice-based formwork allows for autonomous demolding through the melting of the ice after the cast concrete element has cured. This approach demonstrates the potential of ice formwork systems in producing architectural products that are economically, environmentally, and structurally efficient.

While the processes mentioned above can be used to create unique geometries, they are methodologically static, meaning that the outcome is largely predetermined before the forming process even begins. However, this research aims to explore the potential of a materially reactive formwork, drawing inspiration from the dynamic energetics observed in the plant analog, which will be discussed in the next section.

### 3. Framing the Biological Analog

The human impact on the atmosphere is undeniable, and plants offer a tested guidebook for survival in a changing climate. In fact, Velcro, one of the most commonly cited examples of biomimetic design was inspired by the hooks on burrs, the small pods found on grasses used to spread their seeds. A key adaptive feature of most plants species is their ability to adjust observable characteristics in response to changes in the environment, altered dramatically by small changes in external stimuli [20]. Specifically, plants can change their physical appearance, forage for light, water and soil, reproduce, and allocate resources, all from a fixed location [21]. Their innate ability to acclimate to unfavorable and variable conditions makes them a compelling analog for creating a more resilient built environment and there are many researchers exploring these possibilities.

Serving as a bridge between earth and atmosphere, plants release oxygen and water as a byproduct of photosynthesis, the mechanism by which atmospheric carbon is converted into glucose to fuel plant activity. This energy is then distributed throughout the plant

to support the essential processes needed to sustain plant life. Though photosynthesis is diverse and can occur in many forms, a small percentage of known plant species have evolved to supplement this process with metabolic pathways that are more resource efficient [22]. These physiological adaptations have evolved multiple times independently in many species, a phenomenon known as convergent evolution, which highlights their biological importance and makes them an excellent source of inspiration for climate-responsive design [23].

There are three known types of metabolic pathways found in plants: $C_3$ Photosynthesis, Crassulacean Acid Metabolism (CAM), and $C_4$ Photosynthesis. $C_3$, the most common form of photosynthesis, refers to a three-carbon metabolic process where carbon dioxide ($CO_2$) enters the plant through small pores in the leaf membrane called stomata. However, when stomata open to absorb $CO_2$ and begin photosynthesis, they also expel water that may be essential for other cell functions, leaving $C_3$ plants at a disadvantage during drought or high-heat conditions [21]. This process, known as transpiration, is a series of compromises, either cooling or dehydrating the plant. Particularly in hot climates, a plant may become quickly dehydrated as water departs through transpiration, as these plants have no innate signal to stop photosynthesis during sunlit hours. Instead, the plant wilts to avoid contact with the sun and sheds leaves to prevent transpiration. With no mechanism to pause photosynthesis under extreme conditions, the plant may eventually starve.

Though all plants are capable of $C_3$ photosynthesis, many species have developed a specialized four-carbon ($C_4$) pathway that helps them to overcome the challenges mentioned above. While only a limited number of species are known to use $C_4$ photosynthesis, up to 79% of them are grasses and sedges [24], which are some of the biggest and most widespread families. Found especially in tropical climates, these hardy plants can tolerate higher temperatures and extreme conditions [25]. While the other two pathways, $C_3$ and CAM, function within only one cell type, $C_4$ plants split activity between two cell types to prevent the inefficiencies from stomatal transfer of gases. Carbon is first converted into malic acid before it is converted into glucose, stabilizing metabolic processes during unfavorable conditions. This separates $O_2$ from $CO_2$ so that photosynthesis can proceed even when the stomata are closed to prevent desiccating the plant [26]. This acid prevents the starter enzyme from bonding with $O_2$ when stomata are closed. $C_4$ plants can return to $C_3$ photosynthesis when environmental conditions are more favorable to the plant as signaled by stomatal rhythms and $CO_2$ concentrations. These anatomical and physiological adaptations, as well as the reallocation of resources, make it possible for the plant to open and close its pores depending on conditions both inside and outside the plant, allowing the organism to be productive and water efficient in extreme conditions. Though plant physiology is more complex than described here, the next section will define the specific properties that were used to inform the translation processes used in this research.

### 3.1. Compartmentalization (Morphology) and Energetic Phasing (Physiology)

Because plants are static in nature and must rely on rapid chemical responses to regulate and maintain homeostasis under environmental stress, junctions between cells are pivotal for organism survival. The primary component that makes $C_4$ plants more efficient is the compartmentalization of the cells used to create a metabolite gradient between cell types. In $C_4$ plants, carbon is fluxed into one cell and stored until a gradient is formed. The stored carbon is then moved to a separate cell to be fixed into a four-carbon molecule [27]. The spatial separation of metabolic phases is facilitated by the evolution of Kranz anatomy, a unique cell organization characterized by a concentric layered organization, shown in Figures 2 and 3, where photosynthetic processes are divided into two distinct cell types [28]. $CO_2$ is first absorbed into the mesophyll (PM) cells, where fixation occurs entirely in $C_3$ plants, which have thin walls and large intercellular spaces [29]. In $C_3$ plants, fixation occurs entirely within the mesophyll. But in the more efficient $C_4$ plants, carbon leaves this cell as malic acid and enters the bundle sheath (BS) cells, which have notably thicker walls and very few intercellular spaces. Oxygen cannot enter this cell, so the malic acid

can be efficiently fixed into glucose without risk of photorespiration. This division of labor between different cell types minimizes the risk of photorespiration and maximizes the efficiency of photosynthesis. The pressure against the cell wall forces the guard cells of the stomata to stay open, thereby allowing consistent influx of $CO_2$ [30]. Regular flow of $CO_2$ maintains photosynthesis, which in turn allows the plant to cycle energy, grow, and live. Conversely, when the pressure is reduced, the cells deflate to close the stomata and stop the $CO_2$ flux and reduce photosynthesis. This pausing of photosynthesis is detrimental in preserving an organism's essential resources during stress conditions.

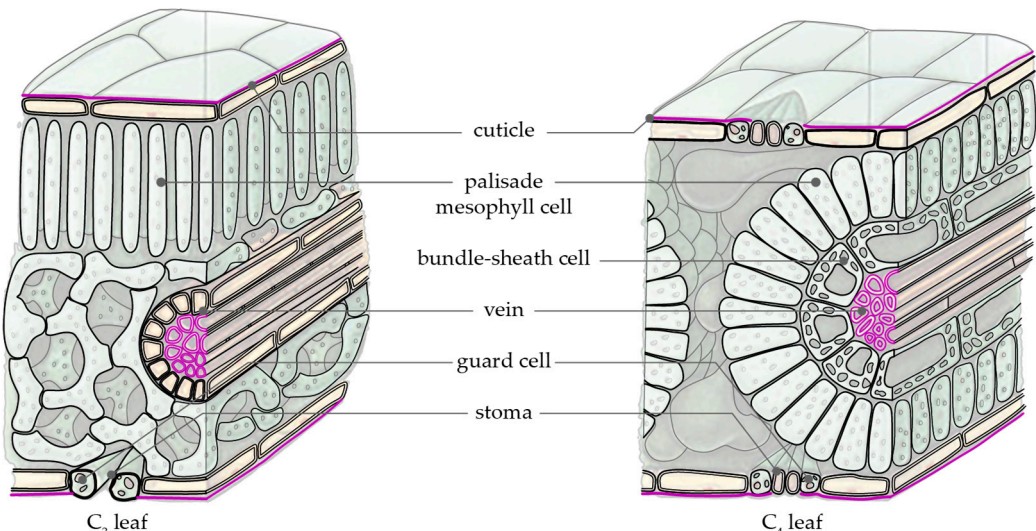

**Figure 2.** $C_3$ (**left**) and $C_4$ (**right**) leaf anatomy. Image adapted from Dr. Darlene Campbell, Cornell University.

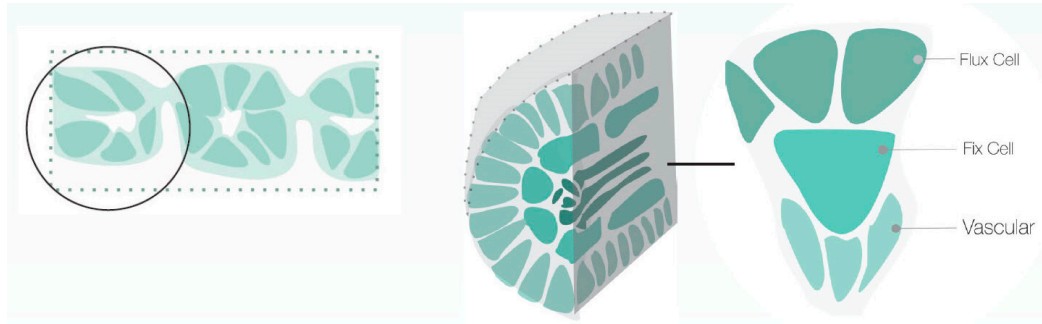

**Figure 3.** Kranz anatomy analysis diagram.

### 3.2. Summary of Relevant Plant Systems

By drawing inspiration from the physiological aspects of plant biology, where cells serve as the fundamental building materials, it is possible to conceive of buildings as metabolic interfaces that engage in a dynamic exchange with the environment. In this project, the incorporation of the principles of compartmentalization and energetic phasing could enhance a building's environmental performance and adaptability to changing climatic conditions. Buildings, much like organisms, can (and often do) employ "smart" materials and systems. These sophisticated technologies can seamlessly adjust energy consumption, ventilation, and temperature control in direct response to fluctuating environmental conditions. Within these adaptive conditions, spaces can organically "breathe" with openings and closures, sensitively responding to sunlight, temperature, and occupancy patterns, creating a dynamic relationship between building occupants and the natural world, blurring the boundaries between indoor and outdoor spaces. In this vision, buildings

cease to be static constructs, instead morphing into living entities that echo the inherent adaptability found in the natural world. However, conventional architectural environments and building materials are static and actuated by advanced mechanized systems. Designed with predetermined behaviors in mind, these systems are optimized for predictability and control through mechanical technologies.

In response, this research explores the plant's dynamic functions to provide insights for construction processes that are themselves dynamic, focusing on methodology instead of predetermined outcomes. By employing materials capable of regulating temperature and energy dynamics within the built environment, we explore a materially autonomous response triggered by environmental cues to inform a climate-responsive design methodology. This approach not only looks to the natural analog for inspiration but attributes creative agency to each material's unique properties, which actively inform and co-create the process of making. The results extend beyond mere dynamic products and assemblies, encompassing dynamic and responsive practices that transcend the conventions of traditional construction. Our research extends beyond dynamic processes to examine how predetermined outcomes can be influenced by the inherent behaviors and properties of materials, exploring the realms of material and design autonomy. The symbiotic relationship between our design process and the biological analog introduces a novel paradigm where the material properties play a central and influential role in shaping the built environment.

## 4. Methodology

Traditional biomimetic translation methods often lean toward literal interpretations, focusing predominately on the end result (outcome) rather than the underlying processes (methodology). However, this research takes a different approach by grounding its methodology in biological principles, allowing the natural analog to guide the process and reduce the influence of human knowledge systems. This approach draws parallels with Menges' Hygroskin Pavilion, which utilizes the innate, responsive qualities of wood and other natural materials, as well as the research on dynamic formwork. However, the key difference is that the climate-responsive behaviors are applied to the fabrication phase instead of the end result. Central to this biomimetic methodology is emulating the plant's dynamic use of resources as an environmental stabilizer. As a result, the core objective of this research is to explore materials that interact and influence each other during fabrication, allowing the outcome to be shaped by the inherent behaviors and energetic processes of materials in transition. For that reason, we were drawn to poured materials, which are dependent on temporary support, or formwork, during construction. In order to achieve an impactful transfer of energy, the receiving medium must respond to this influx of energy so that the final outcome would be derived by the energetic dialogue between the poured and supporting medium.

*Prototype Experiment*

To explore these concepts, the authors embarked on a small-scale experiment using wax and ice. These materials were chosen because they are readily accessible phase-change materials that change states under conditions typical of the built environment. Typically, formwork is constructed in advance of the poured material, but in this approach, they are formed concurrently in order to inform one another, mimicking the energetic phasing of the $C_4$-photosynthesizing plant. In this experimental investigation, shown in Figure 4, we aimed to understand the intricate process of energetic exchange between curing wax and melting ice, exploring whether these interactions could lead to unique architectural outcomes.

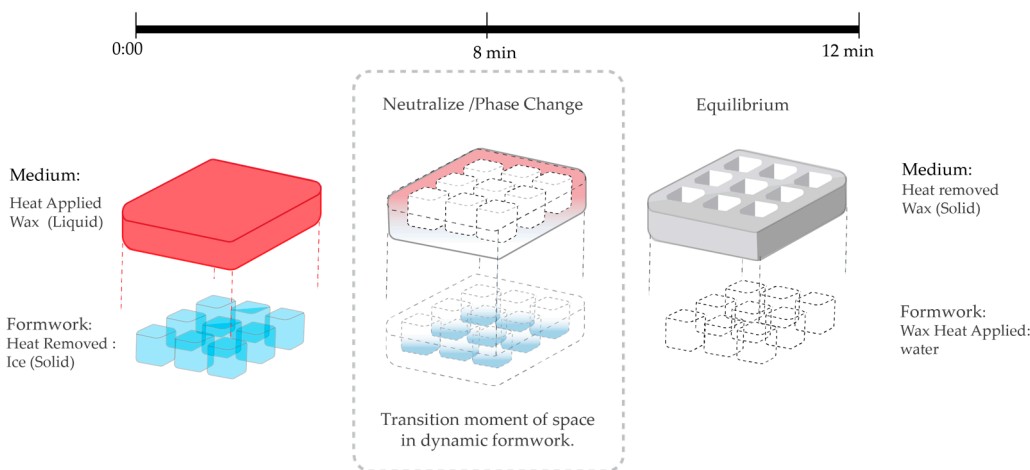

**Figure 4.** Dynamic formwork phase change analysis diagram.

Experimental materials are shown in Figure 5. First, hot wax is poured into a tray filled with ice cubes of varying shapes and sizes. The settling of ice aggregates of different sizes, each melting at its own rate, contributed to the unevenness and fragility in certain areas of the cured wax. Multiple iterations of this process were performed, employing different techniques to manipulate the ice cubes' movement and orientation within the wax. Figure 6 (left) shows an approach to layering using multiple ice dimensions and wax pours. This deliberate variation in approach resulted in a unique and unpredictable series of air pockets forming within the cured wax that could manipulate airflow through the material, as shown in Figure 6 (right). This outcome underscored the dynamic and evolving nature of the process, reminiscent of the adaptive responses observed in plant systems when faced with changing environmental conditions.

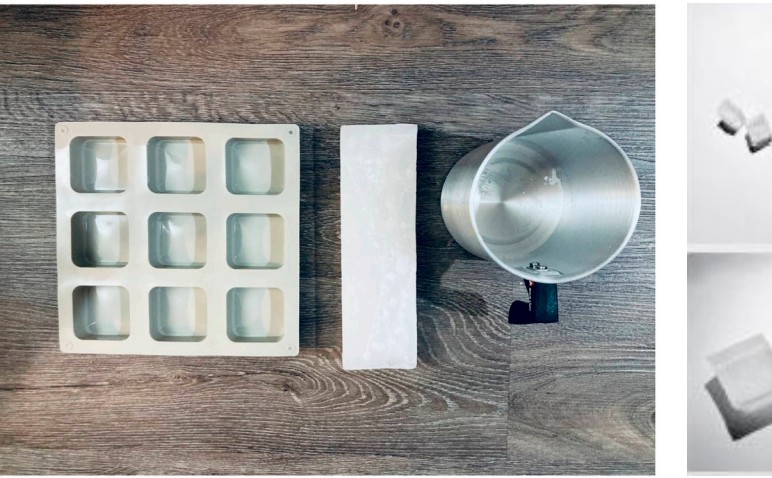
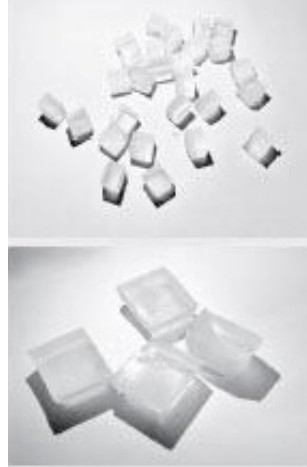

**Figure 5.** Experimental setup showing flexible tray, solid wax, and melting pot. Ice cubes shown on right.

Just as plants adapt to shifting environmental conditions, our approach aimed to create a dynamic architectural response that could adapt to the evolving needs of users and the prevailing environmental conditions. The dynamic response was characterized by the wax material's ability to undergo structural transformations in response to changing energy inputs from the melting ice. Additionally, one of the fundamental principles explored in this methodology was the idea of material autonomy. We sought to investigate whether the cured wax could autonomously respond to environmental stimuli without human intervention. Material integrity, in this context, was not a static attribute but a dynamic quality inspired by the resilience of plant systems. Our goal was to create a material system

capable of responding to environmental challenges through adaptive strategies, similar to the way plants adapt to changes in their surroundings.

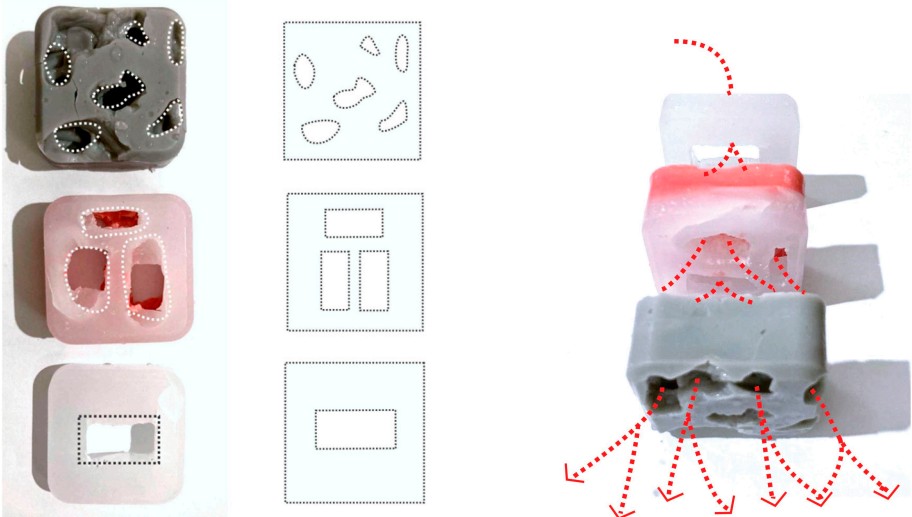

**Figure 6.** (**left**) Showing the layers from various ice shapes. (**right**) Diagram of how air would filter through the layered material.

## 5. Results

The process of wax curing and ice melting involves energy exchange, driven by the temperature differences between the two materials, as well as the stored (latent) energy within the different phases of the two mediums. When hot wax is poured onto ice, it undergoes a phase change from liquid to a solid state. This phase transition is exothermic, meaning it releases energy in the form of heat. Conversely, the ice cubes absorb this heat energy, causing them to melt and transition from a solid to a liquid state. The energy exchange during the phase transition is crucial in determining the final architectural outcome. In an attempt to emulate the energetic behaviors observed in $C_4$-photosynthesizing plants, two dynamic materials were paired through concurrent phase changes, albeit in opposite directions. This intriguing dynamic was created due to the heat generated by the hot wax, which accelerated the melting of the ice, facilitating an exchange of heat energy between the two materials. The outcome of this process resulted in the emergence of new and indeterminate qualities, significantly influencing the formation of pores, the spatial configuration, and the behavior of the resultant materials.

During the fabrication process, we aimed to establish continuity between the poured medium (wax) and the formwork (ice), creating a seamless integration of space and dynamic processes akin to the principles underlying $C_4$ photosynthesis. The use of distinct materials—ice and wax—offered a distinctive relationship between the medium and the formwork, primarily due to their inherent ability to transition between liquid and solid states. The interaction between the hot wax and the ice cubes is central to the formation of these unique qualities. As the ice melts, it effectively harnesses the stored heat energy from the melted wax. This energy transfer not only facilitates the ice's transformation from a solid to a liquid state, but also plays a pivotal role in determining the final volume and structure of the resulting material, as shown in Figures 7 and 8. Especially, in Figure 7, it is most evident that the form starts to break down as the ice cube melts, starting with rigid corners and transitioning to smaller, more fluid geometries. Instead of just releasing that energy back into the atmosphere, it is used to manipulate and inform the making process in the same way that C4-photosynthesizing plants manage resources.

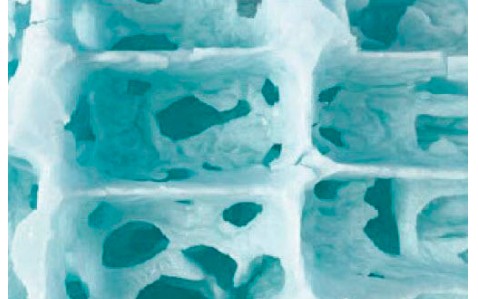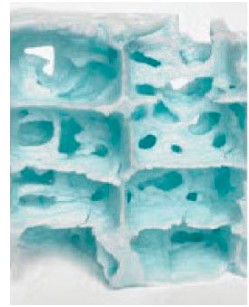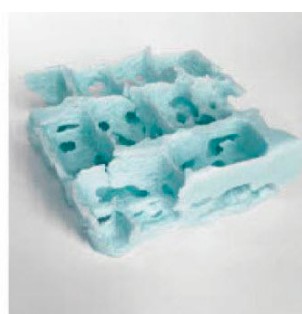

**Figure 7.** Wax prototypes with continuous pores created by melting ice.

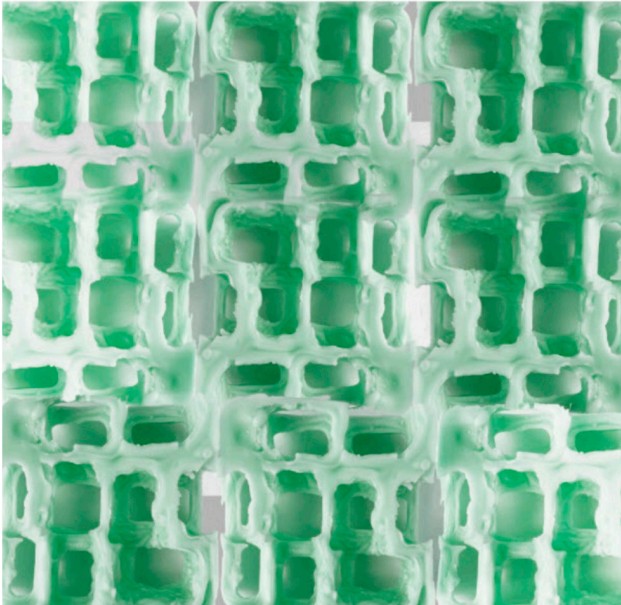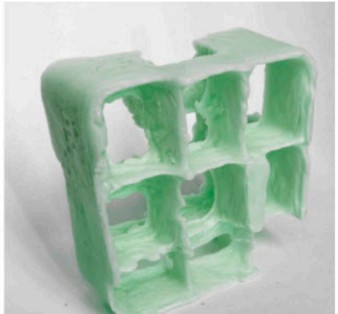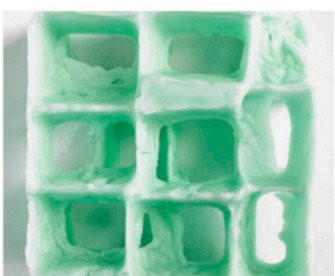

**Figure 8.** Results from cubic ice forms.

While the present study primarily remains a conceptual exercise, its findings suggest promising avenues for the future development of practical materials with applications in the construction industry. Further research in this domain is both warranted and encouraged, as the intersection of these materials offers a captivating prospect for delving deeper into the intricate mechanisms governing natural systems. This research paves the way for further exploration and potential real-world applications in material science and architecture, showcasing the innovative possibilities inherent in biomimetic design and construction.

## 6. Discussion and Conclusions

In this paper, it is argued that non-linear fabrication processes, or those with a high level of variance instead of deterministic outcomes, can generate novel, bio-inspired forms. This project aims to leverage the natural, physiological adaptations of plants that enable their survival in challenging environments. To achieve this, the thermal properties of conventional building materials were investigated, enabling them to respond to environmental conditions independently, without requiring advanced technology or human intervention.

The outcomes of this study demonstrate the feasibility and effectiveness of using climate-responsive, and naturally informed dynamic formwork applications, offering unique and promising solutions for sustainable architectural practices. The findings contribute to the existing body of research on sustainable and dynamic formwork systems, expanding knowledge on the potential use of unconventional and natural materials in the construction industry. Aside from the benefits of using ice and wax as recyclable, clean, and cost-effective formwork, the dynamic interaction between materials in transition presents

a unique opportunity for further bio-inspired design methodologies. The transfer of energy (heat) between the two materials results in a final form, and the fabrication process prioritizes material sensitivity over strict adherence to order and determinacy, leading to irregular and unpredictable geometries.

The methodology employed in this work involved a detailed exploration of the energetic exchange between curing wax and melting ice, considering the exothermic processes of the phase transition and how this energy influenced the final architectural outcomes. The iterative and evolutionary approach celebrated material autonomy and environmental responsiveness, drawing inspiration from the adaptive strategies observed in plant systems. The focus is on the process, rather than the final product, to provide a foundation for future investigations in architectural explorations of dynamic formwork. The ultimate aim is to develop a novel approach that can be extended to various architectural materials, such as wood and concrete, opening up new possibilities for creating intricate and unconventional shapes that were previously unattainable using traditional formwork techniques. Ice and wax, or other inherently reactive (dynamic) materials can be utilized as a dynamic formwork in the architectural fabrication processes. The fabrication process can be adjusted by manipulating these variables, such as the size or shape of the ice cubes used in this exercise, allowing for endless possibilities in the design outcome. This research is focused on exploring the potential of this material combination and is not limited to a specific end result.

Viewed as a living organism, the built environment consists of fluid, ever-changing human-made ecosystems that respond to the dynamic processes of urban life. The modern era has ushered in a new level of "dynamic", climate-responsive architecture. Modern architectural technologies, design, and fabrication techniques are being developed with the help of biomimicry and nature-inspired concepts. Through the collaboration of biologists, ecologists, and designers, biomimicry has the potential to escape its infancy stage to become a reliable form of research. Although we may never fully comprehend the complexity of nature, we can acknowledge its inherent wisdom and grant it authorship in the creation of a climate-responsive built environment. This transformative perspective underscores the synergy between human ingenuity and the innate intelligence of the natural world, offering a promising path towards sustainable design.

**Author Contributions:** Conceptualization, E.L.M. and E.A.C.; methodology, E.L.M. and M.E.; formal analysis, E.L.M. and E.A.C.; investigation, E.L.M., M.E., M.R. and L.S.; resources, E.L.M.; writing—original draft preparation, E.L.M.; writing—review and editing, E.A.C.; visualization, M.E.; supervision, E.L.M. and E.A.C.; project administration, E.L.M. and E.A.C.; funding acquisition, E.L.M. and E.A.C. All authors have read and agreed to the published version of the manuscript.

**Funding:** This work was supported, in part, by funds provided by the University of North Carolina at Charlotte, Faculty Research Grant (FRG-2020-2021).

**Institutional Review Board Statement:** Not applicable.

**Informed Consent Statement:** Not applicable.

**Data Availability Statement:** No new data were created or analyzed in this study. Data sharing is not applicable to this article.

**Conflicts of Interest:** The authors declare no conflict of interest. The funders had no role in the design of the study; in the collection, analyses, or interpretation of data; in the writing of the manuscript; or in the decision to publish the results.

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
