# Peer review of "Process-Oriented Design Methodologies Inspired by Tropical Plants"

_sustainability, doi:10.3390/su152216119_

Round 1
Reviewer 1 Report
Comments and Suggestions for Authors
In this paper, authors investigate the potential of dynamic processes for architectural systems, from the percepstive of adaptive plant physiologyand climate-responsive methodology. The research topic and idea are interesting and attractive, with emphasis on dynamic formwork in building related sectors. Some suggestions for further improvement are as follows.
1. Re-orgainze the whole manuscript in a more scientific style, with clear seperation on research objective, method, results and discussion. For instance, re-compile and integrate the introduction and background parts.
2. The abstract lacks clarification of research novelty and key findings. Instead, the present form pays much expressions on backgrounds.
3. Why tropical plants are chosen as illustartive example? How about others, since climatic adaption might go for each species.
4. It is highly suggested to make more quantitative analysis, rather than qualitative expressions in the present form.
5. Extend the litertuare review with updated citations, especially journal articles, in Architectural-Climatic-Adaptive regards. Most references here were published more than 5 years ago.
6. For building engineering related biological analogy, more detailed statements or examples are necessary.
7. Modelling the phase change materials dynamic processes in building applications, or connect to related available research. Clarify the advance for using the new insights of translation methodology, over existing research approaches.
8. Results and discussion parts need strengthening, with specific quantitative and comparative analysis.
9. Re-structure the conclusion part. Several short bullets are more preferable to list key findings or prospects.
Comments on the Quality of English Language
Moderate language editing
Author Response
- Re-orgainze the whole manuscript in a more scientific style, with clear seperation on research objective, method, results and discussion. For instance, re-compile and integrate the introduction and background parts.
The only section that breaks from the traditional format is section “3. Framing the Biological Analog”. We believe that this section is too robust to be integrated with the background section, therefore would like to keep it in its own section.
We moved the information on dynamic formwork up to the ‘background’ section (from methodology) and generally better separated the background section from the rest of the paper.
- The abstract lacks clarification of research novelty and key findings. Instead, the present form pays much expressions on backgrounds.
We have redrafted the abstract to more succinctly describe the research novelty and key findings. We also shortened it to be under the 200-word limit.
- Why tropical plants are chosen as illustartive example? How about others, since climatic adaption might go for each species.
We mention that we are specifically referencing C4 photosynthesis, which is found most commonly in tropical plant species.
- It is highly suggested to make more quantitative analysis, rather than qualitative expressions in the present form.
We agree that would be useful for further development, but this research is framed more as an exploratory study – posing new modes of working (and thinking) instead of proposing specific (measurable) strategies.
- Extend the litertuare review with updated citations, especially journal articles, in Architectural-Climatic-Adaptive regards. Most references here were published more than 5 years ago.
We added a few more current citations and removed a few that were not as relevant as the others.
- For building engineering related biological analogy, more detailed statements or examples are necessary.
We are not exactly sure what is meant by this statement, but I would point to our response for comment #4.
- Modelling the phase change materials dynamic processes in building applications, or connect to related available research. Clarify the advance for using the new insights of translation methodology, over existing research approaches.
Again, I’m sorry that I’m not exactly sure what this comment is suggesting, but we made a better effort to explain the novelty of this research in the results/discussion section.
- Results and discussion parts need strengthening, with specific quantitative and comparative analysis.
While we are not able to add quantitative results at this time, we did expand the discussion and results section. Because this was just a theoretical exercise testing the theory of the biomimetic translation, we are not prepared to address the type of results you may be looking for. Instead of proposing specific bio-inspired strategies, we’re hoping to provoke new modes of thinking about construction of the built environment.
- Re-structure the conclusion part. Several short bullets are more preferable to list key findings or prospects.
We have restructured the conclusion but opted to keep the text in narrative format to align with the rest of the paper.
Reviewer 2 Report
Comments and Suggestions for Authors
The article concerns a complex study on the possible integration of principles inspired by the behavior of tropical plants in the design and production of building components in order to assign them dynamic behavior in response to different environmental conditions. The investigation explores the potential of biomimicry in new sustainable human technology.
The article focuses on a topic that appears really interesting in terms of originality. The topic is truly original and, as declared by the authors, outlines new directions and perspectives from a technological point of view in architecture. It addresses a specific gap in the field due to the limited number of studies related to this specific topic. It is clear, well-written, and relevant to architectural studies.
The other few studies are aimed at similar topics but not exactly coinciding with the present study which considers the state of the investigations to define innovative lines of development.
However, some aspects should be improved.
First of all, the section relating to the methodological approach is not clearly described; in particular, both in the title and in the conclusions the authors refer to the proposed methodology which, unfortunately, does not show adequate clarity. Authors must declare the sequence of operations carried out to achieve the research objective. The section related to the methodological approach needs a big improvement in terms of clarity, to allow the reproducibility of outcomes
The authors do not declare in the abstract that the proposed methodology is functional for the design of formwork but this omission does not facilitate the understanding of the work until the paragraph in which it is declared.
Another aspect to clarify is the design and production of the formwork. In particular, authors must specify how the scale and dimensions of the formwork can influence its construction.
The conclusions are consistent with the evidence and arguments presented; however, a broad description of the formwork model proposed by the authors is highly recommended in order to explain its usability and application areas. In addition, authors should declare the minimum and maximum size of the formwork.
There is a typo in the caption of Figure 6; in fact, the photo number is 7 when it should be 6.
The contribution should be improved by inserting other formwork figures. A greater number of evocative images could facilitate the comprehension of the paper which is clear and well-written but also proposes truly complex themes, not immediately understandable.
References are appropriate
Author Response
First of all, the section relating to the methodological approach is not clearly described; in particular, both in the title and in the conclusions the authors refer to the proposed methodology which, unfortunately, does not show adequate clarity. Authors must declare the sequence of operations carried out to achieve the research objective. The section related to the methodological approach needs a big improvement in terms of clarity, to allow the reproducibility of outcomes
We have expanded the methodology and results sections with additional text and imagery to help with this point.
The authors do not declare in the abstract that the proposed methodology is functional for the design of formwork but this omission does not facilitate the understanding of the work until the paragraph in which it is declared.
We have redrafted the abstract to more succinctly describe the research novelty and key findings. We also shortened it to be under the 200-word limit.
Another aspect to clarify is the design and production of the formwork. In particular, authors must specify how the scale and dimensions of the formwork can influence its construction.
The conclusions are consistent with the evidence and arguments presented; however, a broad description of the formwork model proposed by the authors is highly recommended in order to explain its usability and application areas. In addition, authors should declare the minimum and maximum size of the formwork.
Because this was just a theoretical exercise testing the theory of the biomimetic translation, we are not prepared to address scale of the formwork. Instead of proposing specific bio-inspired strategies, we’re hoping to provoke new modes of thinking about construction of the built environment.
There is a typo in the caption of Figure 6; in fact, the photo number is 7 when it should be 6.
Thank you, we have fixed this and made sure the figure callouts reference the correct images.
The contribution should be improved by inserting other formwork figures. A greater number of evocative images could facilitate the comprehension of the paper which is clear and well-written but also proposes truly complex themes, not immediately understandable.
We have added some additional photos from the making process.
Reviewer 3 Report
Comments and Suggestions for Authors
In the paper, authors explore the potential of dynamic processes within broader architectural systems, using adaptive plant physiology to inspire novel, climate-responsive methodological practices. In their work, authors focused more on the physiological behaviors of tropical plants to build a methodological prototype.
The current work is very important and promising to develop a new kind of sustainable buildings inventions. In my opinion this paper needs major revision.
Comment (1): The papers cited in the article is somewhat. It is recommended to cite some papers published in the last two or three years.
Comment (2): The importance of the prototype in the imitation of nature is presented with qualitative manner. Please reinforce the methodology with measurable results
Author Response
Comment (1): The papers cited in the article is somewhat. It is recommended to cite some papers published in the last two or three years.
Noted. We have added some more current articles in the background section and eliminated some which were less relevant.
Comment (2): The importance of the prototype in the imitation of nature is presented with qualitative manner. Please reinforce the methodology with measurable results
We agree that quantitative data would be useful for further development, but this research is framed more as an exploratory study – posing new modes of working (and thinking) instead of proposing specific (measurable) strategies. We have expanded the methodology and results sections with additional text and imagery to help with this point.
Round 2
Reviewer 1 Report
Comments and Suggestions for Authors
Authors have made necessary revisions and explanations according to last review comments.
Comments on the Quality of English LanguageMinor language and format editing required
Reviewer 3 Report
Comments and Suggestions for Authors
No comments